## Introduction

Non-infectious central nervous system (CNS) diseases can also have clinical presentations similar to those of infectious causes of encephalitis and should also be considered in the differential diagnosis[1]. Out of the viral forms of encephalitis, Japanese encephalitis (JE), is one of the most prevalent forms throughout the world, affecting mostly children and causes 15,000 deaths annually. Due to lack of suitable diagnostic strategies for Japanese encephalitis virus (JEV) and hence delayed treatment, the mortality rate of this disease is very high. Till date, JE can be diagnosed only with clinical symptoms, and serological examination of JE patients. Therefore, there is an urgent need for the development of effective treatment strategies which may be effective before the viral invasion in the brain and spinal cord. For this, an earlier diagnosis based on reliable biomarkers is essential to identify the status and intensity of JE infection. The extracellular fluid of the brain and spinal cord constitutes around 30 to 40% of the cerebrospinal fluid (CSF). Cerebrospinal fluid thus provides an accessible insight into the brain and hence is an ideal body fluid to examine for signature protein profiles for diagnosis or etiology of CNS-related disorders[2].

## Methods

### Ethics statement

The study samples were obtained from the pediatric and adult medicine wards of King George's Medical University (KGMU) in Lucknow, Uttar Pradesh. KGMU Hospital is a teaching hospital which caters mostly to the poor and severely ill from the city and surrounding districts extending upto Nepal. Ethical approval was obtained from the Institutional Ethics Committee of King George's Medical University (KGMU), Lucknow. Written consent for CSF collection was obtained from the patient's guardian. The experiments were carried out in accordance with the institutional approved guidelines.

### CSF samples

A total of 20 CSF samples were obtained from King George's Medical University, Lucknow for this study, These included 10 subjects each with JE and other forms of acute encephalitis respectively (Table 1). The lumbar puncture was done within 3 hours of admission of the patients and the CSF volume collected was 2–3 ml from each individual. Inclusion criteria for CSF collection in both AE and JE patients were: i) age >3 years but excluding pregnant women, ii) presence of fever with altered sensorium of 7 days or less. Exclusion criteria were: i) a firm alternative etiological diagnosis, ii) some contraindication to drug administration.

**Table 1. Details of the cerebrospinal fluid samples from AES and JE patients.**

| Sample Details | AE (n=10) | JE (n=10) |
|---|---|---|
| Patient's Age | 4.5–50 years | 4–45 years |
| Gender (male: female) | 7:3 | 8:2 |
| Presence of IgM | JE IgM negative | JE IgM positive |

### Enzyme linked immunosorbent assay (ELISA) tests

JE was confirmed by the presence of JE IgM in CSF using IgM Capture ELISA (MAC ELISA) kit developed by the National Institute of Virology (Pune) as per the manufacturer's instructions.

### Protein enrichment

Due to the limited volume of the CSF samples, CSF samples from each experimental group were pooled by mixing equal volumes of each sample. The protein concentration of the samples was then determined by Bradford's method (Bradford Protein Assay Kit, Bio-Rad) according to the manufacturer's instructions. The respective CSF samples were then supplemented with 0.1% Triton X100 and protease inhibitors (Sigma Aldrich) and then filtered using a 0.45μm syringe filter to remove particulate matter, concentrated until the desired volume for protein enrichment was reached using a Freeze dryer (Martin Christ, Germany) and subjected to protein enrichment using Bio-Rad ProteoMiner™ Protein Enrichment Kit following the manufacturer's instructions to normalize the levels of high abundance proteins in the CSF.

### Sample preparation and proteomic analysis

The enriched samples were then cleaned using 2D-clean up kit (Bio-Rad Laboratories, CA, US) following the manufacturer's instruction to remove impurities such as nucleic acids, lipids, and salts. 2-DE was performed as described earlier[3]. Briefly, the cleaned up protein pellet was resuspended in sample rehydration buffer and for the first dimension, isoelectric focusing was performed using immobilized pH gradient (IPG) strips (Bio-Rad, USA) of 7 cm size with a pH range from 3.0–10.0 on a Protean i12TM IEF Cell (Bio-Rad, USA). For the second dimension, the proteins were separated by Tris glycine SDS-PAGE on 12% poly acrylamide gel by the method of Blackshear[4]. The protein spots were visualized by overnight staining with Brilliant Blue R-250, destained with $H_2O$, methanol, and acetic acid in a ratio of 50/40/10 (v/v/v) and scanned using Licor Odyssey Infra-red Imaging System. The protein spots visualized exclusively in the JE CSF proteome were excised and stored at -80°C and processed later for in gel trypsin digestion as previously described[5]. The peptides extracted were analyzed by Matrix-assisted laser desorption/ionization-time of flight (MALDI-TOF/TOF) mass spectrometry using AB SCIEX TOF/TOFTM 5800 System. Acquired combined MS and MS/MS spectra were analyzed with ProteinPilot 4.0 Software using MASCOT v 2.3.02 search engine from matrix sciences against the taxonomy *Homo sapiens*. The peak list was searched against the taxonomy *Homo sapiens* at protein sequence Database: UniProtKB-SwissProt sprot_2014-04-16 (544996 sequences; 193815432 residues) Search parameters were as follows: Digestion: trypsin with one missed cleavage; Fixed modification: carbamidomethyl (c); variable modification: oxidation (m); peptide mass tolerance: 100ppm for precursor ion and 0.8 Da for fragment ion with +1 charge state; instrument: MALDI-TOF-TOF.

### CSF cytokine analysis

IL-8, IL-1β, IL-6, IL-10, TNFα, and IL-12 concentrations were measured by flow cytometry using a Human Inflammatory Cytokine CBA Kit, BD Cytometric Bead Array (a kind gift from Dr. Pankaj

Seth, NBRC) (BD Biosciences, San Diego, CA, USA) as per the manufacturer's instructions (n=3 for each experimental group).

### Protein Interaction Analysis

The associations of Vitamin D binding protein were explored using the STRING v10 clustering tool (http://string-db.org/). The confidence score was set at the highest level (0.900) and additional 20 nodes which were indirectly interacting with DBP were asked to show by the software.

## Results

**Dataset 1. Raw data of Cytokine Bead Analysis (CBA) of AE and JE infected CSF sample**

**http://dx.doi.org/10.5256/f1000research.6801.d89533**

The datasheet contains the cytokine profile of three patients of AE and JE CSF samples.

**Dataset 2. PDF version of the MALDI-TOF raw Data of the collected spots of JE CSF**

**http://dx.doi.org/10.5256/f1000research.6801.d89605**

Each folder consists of two PDF files, one file contains the Mascot Search Result and the other file explains the identification of the protein.

### Identification of potential biomarkers for JE in the CSF

The specific JE associated proteins were identified by proteomic comparison of CSF from JEV-infected patients and patients with other forms of encephalitis. Around 16 proteins were found to be exclusively present in the JEV CSF proteome out of which 10 spots could be successfully identified (Figure 1). The observed MW and pI values of the protein spots on the 2- DE gels were compared with the theoretical MW and pI values of corresponding proteins (Table 2). The proteins identified were predominantly DBP,

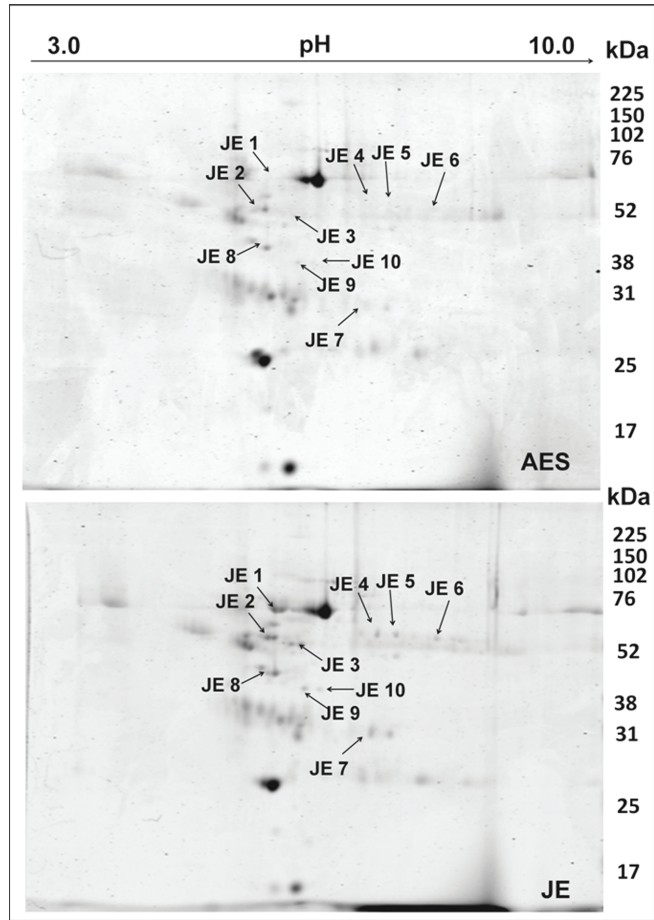

**Figure 1. Comparative proteomic analysis of cerebrospinal fluid from AES and JEV patients.** Cerebrospinal fluid samples were pooled and proteins were extracted and separated on immobilized linear pH gradient IPG strips (pH 3.0–10.0) and then in the second dimension on 12% SDS-PAGE. Spots exclusively visualized in the JE- CSF were marked and excised, and identified by MALDITOF/MS and database searches. The spots are labeled on the gel according to the numbers presented in Table 2. Images are representative of 4 replicate experiments.

**Table 2. Proteins exclusively visualized in the CSF of JEV infected patients, identified by MS/MS analysis of excised spots.**

| Sr. No | Spot No. | Protein ID. (Accession)[a] | Matched peptides | Ion score | % coverage | Mowse Score[b] | MW (theor/ obs) | pI (theor/ obs) |
|---|---|---|---|---|---|---|---|---|
| 1. | JE-1 | Serum albumin (NP_000468) | K.LVNEVTEFAK.T<br>K.SLHTLFGDKLCTVATLR.E<br>K.KYLYEIAR.R<br>K.YLYEIAR.R | 30<br>56<br>56<br>44 | 59 | 1480 | 71/70 | 5.9/5.5 |
| 2. | JE-2 | Vitamin D-binding protein (NP_000574) | R.KFPSGTFEQVSQLVK.E<br>K.EVVSLTEACCAEGADPDCYDTR.T<br>K.SCESNSPFPVHPGTAECCTKEGLER.K<br>K.HQPQEFPTYVEPTNDEICEAFR.K<br>K.HLSLLTTLSNR.V | 77<br>89<br>142<br>74<br>97 | 31 | 614 | 54/60 | 5.4/5.4 |
| 3. | JE-3 | Fibrinogen gamma chain (NP_000500) | R.DNCCILDER.F<br>R.YLQEIYNSNNQK.I<br>K.QSGLYFIKPLK.A<br>K.IHLISTQSAIPYALR.V<br>K.IHLISTQSAIPYALR.V | 34<br>72<br>50<br>90<br>36 | 24 | 595 | 52/54 | 5.37/5.9 |
| 4. | JE-4 | Fibrinogen beta chain (NP_005132) | R.GHRPLDKK.R<br>K.HQLYIDETVNSNIPTNLR.V<br>R.SILENLR.S<br>R.TPCTVSCNIPVVSGKECEEIIR.K<br>R.QDGSVDFGR.K | 22<br>69<br>21<br>49<br>40 | 35 | 286 | 56/60 | 8.54/6.9 |
| 5. | JE-5 | Fibrinogen beta chain (NP_005132) | R.EEAPSLRPAPPPISGGGYR.A<br>K.HQLYIDETVNSNIPTNLR.V<br>R.SILENLR.S<br>R.QDGSVDFGR.K | 19<br>78<br>26<br>23 | 22 | 199 | 56/60 | 8.54/7.2 |
| 6. | JE-6 | Fibrinogen beta chain (NP_005132) | R.GHRPLDK.K<br>R.EEAPSLRPAPPPISGGGYR.A<br>K.HQLYIDETVNSNIPTNLR.V<br>R.SILENLR.S<br>R.TPCTVSCNIPVVSGKECEEIIR.K | 10<br>19<br>68<br>26<br>42 | 25 | 232 | 56/54 | 8.54/7.7 |
| 7. | JE-7 | Complement C4b (NP_001002029) | R.EAPKVVEEQESR.V<br>R.VHYTVCIWR.N<br>R.YVSHFETEGPHVLLYFDSVPTSR.E<br>R.ECVGFEAVQEVPVGLVQPASATLYDYYNPERR.C<br>R.RCSVFYGAPSK.S | 88<br>61<br>115<br>55<br>18 | 17 | 864 | 194/31 | 6.8/6.8 |
| 8. | JE-8 | Actin, cytoplasmic-1 (NP_001092) | K.AGFAGDDAPR.A<br>R.AVFPSIVGRPR.H<br>K.DSYVGDEAQSKR.G<br>K.DSYVGDEAQSKR.G | 33<br>53<br>52<br>120 | 48 | 1070 | 42/50 | 5.29/5.4 |
| 9. | JE-9 | Complement C4b (NP_001002029) | R.EAPKVVEEQESR.V<br>R.VHYTVCIWR.N<br>R.CSVFYGAPSK.S<br>R.LLATLCSAEVCQCAEGKCPR.Q<br>R.GLQDEDGYR.M | 88<br>60<br>54<br>142<br>41 | 12 | 634 | 194/45 | 6.8/6.0 |
| 10. | JE-10 | Complement C3 (NP_000055) | K.VYAYYNLEESCTR.F<br>K.ACEPGVDYVYKTR.L<br>K.SGSDEVQVGQQR.T<br>K.SDDKVTLEER.L | 90<br>54<br>49<br>34 | 11 | 334 | 188/45 | 6.02/4.8 |

[a]NCBI accession number of identified proteins is mentioned.

[b]MS/MS data of 3 peptides for each spot was searched against NCBI database in the taxonomy group of *Homo sapiens* using Mascot tool.

fibrinogen gamma chain, fibrinogen beta chain, complement C4-B, complement C3 and cytoplasmic actin. Most of the identified proteins were found to be members of the albumin multigene family.

The levels of two pro-inflammatory cytokines IL-1β and TNFα were found to be significantly elevated in JE patients as compared to AES patients while there was no remarkable change in the rest

(Figure 2). The data were analyzed by Student's t-test and a statistical $p$ value<0.05 were considered significant.

The STRING v10 clustering tool was used to explore currently known associations of DBP (GC in Figure 3) and out of all the predicted functional partners, low density lipoprotein related protein-2 (LRP2) which is also known as megalin was found to have the highest interaction score (Figure 3).

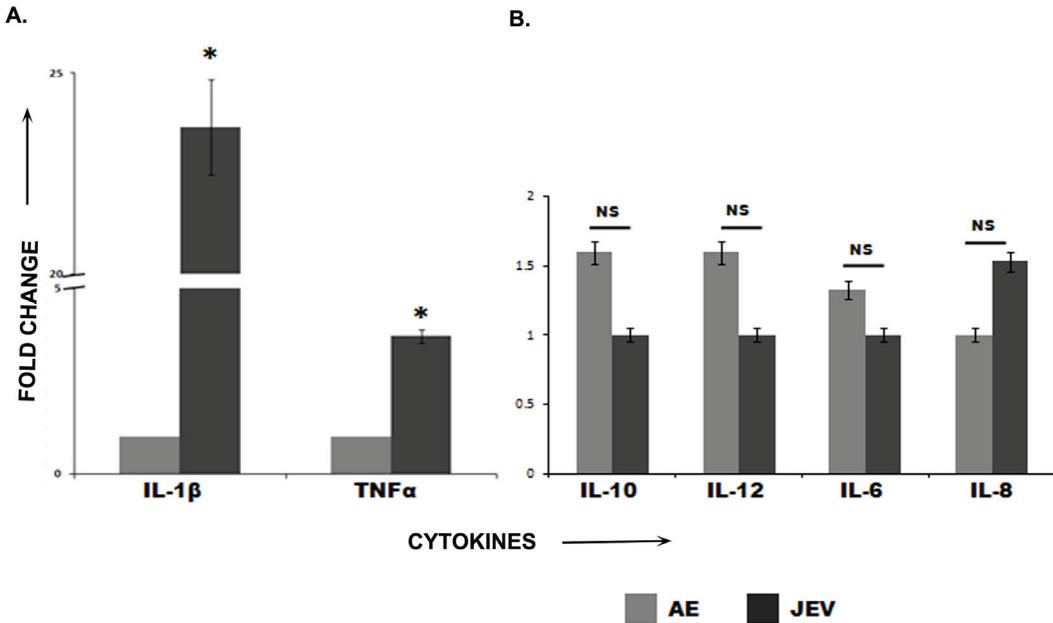

**Figure 2. (A and B). Inflammatory cytokine profile of cerebrospinal fluid from AES and JEV patients.** IL-8, IL-1β, IL-6, IL-10, TNFα, and IL-12 concentrations were measured by flow cytometry using Human Inflammatory Cytokine Kit (BD Biosciences, San Diego, CA, USA) as per manufacturer's instructions. Elevated levels of IL-1β and TNFα were observed in the JE samples (**2A**) whereas no significant changes were observed in the rest (**2B**). The image is a representative of 3 replicate experiments.

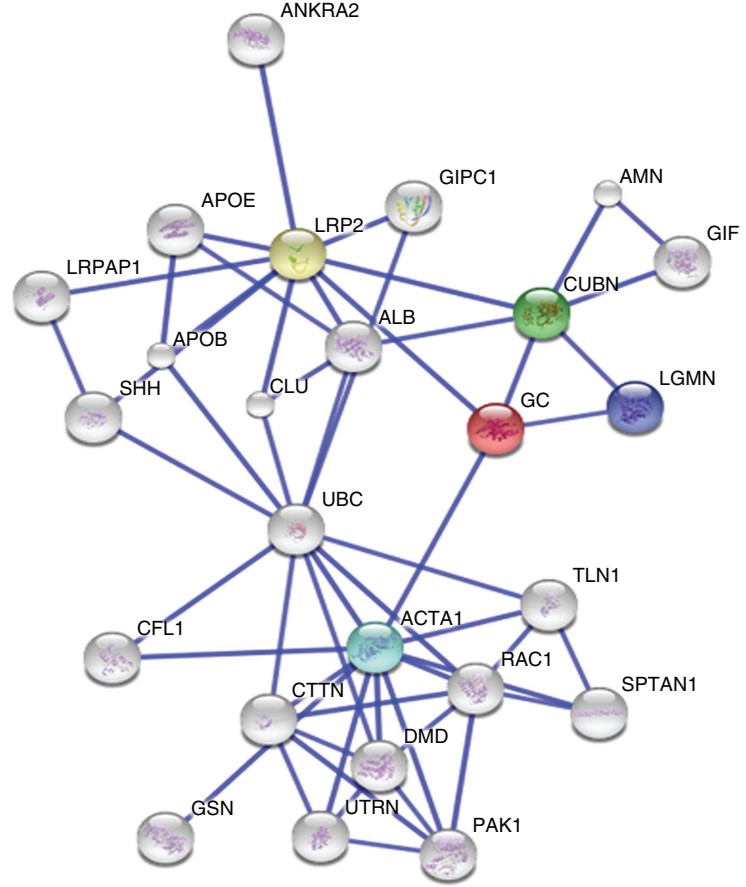

**Figure 3. The associations of Vitamin D binding protein ('GC' in the figure) were explored using the STRING v10 clustering tool.** The colored nodes signify a direct interaction with DBP whereas the white nodes denote a distant interaction. 4 proteins, LRP2 (yellow node), Cubilin (CUBN, green node), alpha 1 actin (ACTA1, blue node) and legumain (LGMN, violet node) are directly interacting with DBP and LRP2 is being shown the highest interacting score (0.983).

## Discussion

JE is one of the most dreaded forms of epidemic and sporadic encephalitis in the tropical regions of Asia with a very high mortality rate especially among children and young adults. The CSF proteome has been explored extensively for the identification of biomarkers for diseases like Alzheimer's disease, brain tumors, multiple sclerosis[6] but until now, few studies have investigated the association between imbalance of CSF elements and severity of JE infection. Recent advancements in proteomic approaches by mass spectrometry have aided the identification of useful predictive biomarkers. The present study provides the first insight into the potential CSF biomarkers associated with JEV neuroinvasion using proteomic methodologies. The CSF proteome from JEV infected individuals was compared to that of AES patients by 2DE-MS based approach. These experiments successfully identified a set of proteins with abnormal expression patterns in the CSF of JE patients which included 6 major proteins belonging to the albumin multigene family. One of the most interesting findings was the DBP which is an abundant multifunctional protein with roles in vitamin D metabolite transport, actin sequestration, and regulation of immune responses[7]. In a recent study, Yang *et al.*[8] reported DBP to be a potential diagnostic biomarker for the progression of multiple sclerosis where this protein has been shown to interact with actin and play an important role in removal of excess actin. In addition elevated levels of DBP were also shown to be adverse to recovery. Interestingly, we have also observed an increase in the levels of cytoplasmic actin in the CSF of JE patients as opposed to that of patients with AES. Elevated levels of cytoplasmic actin in the CSF have been previously related to severe axonal degeneration associated with progressive multiple sclerosis[9]. Hence the abnormal levels of the aforementioned interacting proteins in case of JE may be attributed to the extensive neurodegeneration taking place in the CNS. Another important finding was the abnormal increase in the levels of complement proteins C3 and C4b. It has been previously suggested that the activation of the complement system is involved in the pathogenesis of several neurodegenerative diseases like Alzheimer's disease, Parkinson's disease, and multiple sclerosis[10]. A recent study has also shown that complement C3 is causally involved in the inflammatory neurodegeneration[11]. Some studies have also shown that DBP can enhance complement component 5a-mediated macrophage chemotaxis via binding to C5a[12]. We also detected elevated levels of fibrinogen beta and gamma chains as well as serum albumin in the CSF of JE patients. Fibrinogen and serum albumin are normally excluded from the brain by the blood brain barrier (BBB) and their presence in the CSF is an obvious sign of the disruption of the BBB during JE infection. A recent study has shown that fibrinogen leakage upon BBB disruption triggers perivascular microglial clustering in turn causing neuroinflammation and subsequent axonal damage[13]. Hence the accumulation of fibrinogen in the cerebral parenchyma may play a critical role in neuroinflammation and JE pathogenesis.

In our study, DBP, complement proteins C3 and C4b and fibrinogen beta and gamma chain were found to be increased in expression in JE patients when compared to AES patients. Elevated levels of proinflammatory cytokines like IL-1β and TNFα were indicative of peripheral immune activation subsequently leading to an up-regulation of CNS cytokine production[14]. Since DBP has been an important finding of this study, an attempt was made to explore the interacting partners of this protein in order to decipher its possible role in JEV neuropathogenesis using STRINGv10 cluster analysis tool[15]. LRP-2 or megalin was found to have the highest interaction score. LRP-2 belongs to a group of surface endocytic receptors, which bind and internalize extracellular ligands which also include DBP and is known to play a key role in the clearance and entrance of many proteins from the brain or CSF[16]. These findings highlight a severe disruption of the BBB during JE infection ultimately leading to subsequent pathogenesis and neurodegeneration and DBP may have an important role as a potential JEV biomarker which can be established with further studies.

## Conclusion

We can suggest that DBP may be an important marker for JE. Further studies need to be carried out to further validate this putative biomarker.

## Data availability

*F1000Research:* Dataset 1. Raw data of Cytokine Bead Analysis (CBA) of AE and JE infected CSF samples. The datasheet contains the cytokine profile of three patients of AE and JE CSF samples. 10.5256/f1000research.6801.d89533[17]

*F1000Research:* Dataset 2. PDF version of the MALDI-TOF raw Data of the collected spots of JE CSF.

Each folder consists of two PDF files, one file contains the Mascot Search Result and the other file explains the identification of the protein. 10.5256/f1000research.6801.d89605[18]

### Author contributions

AB, NS and SM designed the experiments. NS and SM performed the experiments. AB and NS wrote the main manuscript and prepared the figures. RK and PT collected the samples. Mass Spectrometry was performed by AS. All authors reviewed the manuscript.

### Competing interests

The authors declare no competing interests.

### Grant information

AB is awarded the Tata Innovation Fellowship (ABS/DBT/0515/062) from the Department of Biotechnology. The project is also funded by National Brain Research Centre core funding. RK is funded by a grant from NBRC.

*I confirm that the funders had no role in study design, data collection and analysis, decision to publish, or preparation of the manuscript.*

### Acknowledgements

The authors would like to acknowledge the kind help provided by R. Rajendra Kumar Reddy, Central Proteomics Facility at Institute of Life Sciences, Bhubaneswar, India.

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

**Data Source**

