## [Reviewer Report]

Sengupta
*et al. *submitted a research article reporting novel CSF biomarkers of Japanese encephalitis (JE). JE is a leading cause of viral encephalitis in Asia. However, there are still many gaps in our understanding of the pathogenesis of JE in humans. Early diagnosis based on reliable biomarkers is essential to identify the status and intensity of JE infection. Here, the authors compared CSF proteomes from ten patients with JE and ten patients with another form of acute encephalitis by 2D gel electrophoresis followed by mass spectrometry. The analysis identified elevated levels of vitamin D-binding protein (DBP), fibrinogen gamma chain, fibrinogen beta chain, complement C4-B, complement C3 and cytoplasmic actin in CSF from JE patients. This indicates blood-brain barrier disruption, and the DBP also represents a novel and mechanistically important CSF biomarker of JE. The exact role of DBP in the development of JE will be investigated is further studies.

The manuscript is well written, easy to follow, and provides very important data to all interested in the mechanisms of pathogenesis of neurotropic flaviviral infections. I fully recommend accepting the manuscript for indexation.   

Minor comments:
I would be interested to know more details on the other forms of the acute encephalitis involved in the study. In the next studies, it would be interesting to compare different forms of encephalitis more specifically, like JE vs. TBE, JE vs. herpetic encephalitis, etc.In the next studies, it would be useful to investigate the biomarker levels in individual patients or to make more pools for each group. In case of one pool investigated, it is not fully clear if the levels of biomarkers are increased in all or most of the patients or if there is just only a strong production in one patient. 

I have read this submission. I believe that I have an appropriate level of expertise to confirm that it is of an acceptable scientific standard.

---

## [Reviewer Report]

This paper is a significant addition to our knowledge in JE encephalitis and have identified markers using proteomic approach that would also suggest damage to the blood brain barrier. The experiments carried out support the conclusions. I recommend indexation. The DBP as marker will need more JE and non-JE cases to confirm - so they must indicate that possibility.

I have read this submission. I believe that I have an appropriate level of expertise to confirm that it is of an acceptable scientific standard.

---

## [Reviewer Report]

The manuscript entitled “Cerebrospinal Fluid Biomarkers of Japanese Encephalitis” by Sengupta and colleagues reports their novel findings unique biomarkers for an important disease wherein encephalitis caused by Japanese Encephalitis Virus causes almost 15,000 deaths every year. Currently the JEV infection can be diagnosed only by using serology and clinical symptoms. Therefore there is an urgent need to identify biomarker(s) for this disease. They used cerebrospinal fluid from 10 JE and 10 AE patients and performed 2D and mass spectrometry on these samples. They observed 6 different proteins that were significantly increased in JE patients. They also found that IL-1β and TNF-α were also significantly elevated in JE patients. Their further analysis indicated DBP as potential biomarker in JE patients.

Both the Tables and 3 figures are necessary and results in these items support the conclusions very well stated in the manuscript. This manuscript would be extremely useful to the people involved in the general area of JE research and particularly to those involved in diagnostics. The manuscript is well written and this study provides a very important and significant advancement in the field.

I have read this submission. I believe that I have an appropriate level of expertise to confirm that it is of an acceptable scientific standard.